# Intention to Purchase Halal Cosmetics: Do Males and Females Differ? A Multigroup Analysis

Abdul Hafaz Ngah [1,*], Serge Gabarre [2], Heesup Han [3], Samar Rahi [4], Jassim Ahmad Al-Gasawneh [5] and Su-hyun Park [6]

1 Faculty of Business, Economy and Social Development, Universiti Malaysia Terengganu, Kuala Nerus 21030, Malaysia
2 College of Arts and Sciences, University of Nizwa, Ad Dakhiliyah, Birkat Al Mouz, Nizwa PC 616, Oman; sergegabarre@unizwa.edu.om
3 College of Hospitality and Tourism Management, Sejong University, Seoul 05006, Korea; heesup.han@gmail.com
4 Hailey College of Banking and Finance, University of the Punjab, Lahore 54000, Pakistan; sr_adroitl@yahoo.com
5 Department of Marketing, Digital Marketing, Faculty of Business, Applied Science Private University, Amman 11931, Jordan; j_algasawneh@asu.edu.jo
6 Department of Hotel and Leisure Management, Pai Chai University, Daejeon 35345, Korea; kayla9252@gmail.com
* Correspondence: hafaz.ngah@umt.edu.my; Tel.: +609-6884573

**Abstract:** As Muslims bound to Islamic teachings, the attitude of young millennials preferring non-*halal* international cosmetics is trivial. Despite the acceptance of halal food, literature on the acceptance of halal cosmetics remains scarce. The intention to purchase halal cosmetics is crucial for the sustainability of halal cosmetics manufacturers. The authors used the theory of planned behavior to identify factors influencing the purchase intention of halal cosmetics among Muslim millennials. Since cosmetics are not exclusively used by females, as males are starting to use them in their daily lives, gender was incorporated into the framework to assess its moderating effect on the relationship. Furthermore, brand image was included in the theory of planned behavior. Data were collected from three universities in Malaysia. A total of 501 responses were analyzed with smart partial least squares to run a multigroup analysis. The analysis revealed that subjective norms have a stronger effect on females, and perceived behavioral control has a greater effect on males. Although attitude and brand image have a positive effect on the intention to purchase halal cosmetics, gender has no effect. The findings are essential for halal cosmetics manufacturers to craft a marketing strategy aimed at Muslim millennials in Malaysia.

**Keywords:** intention to purchase; halal cosmetics; multi-group analysis; millennials



## 1. Introduction

For Muslims, consuming halal products is not an option but an obligation. Owing to an increase in understanding of religious obligations, the demand for halal products has demonstrated the fastest growth in the world market [1]. Halal products are central to Muslims yet the majority of the manufacturers are non-Muslims [2]. The concept of halal is not only limited to food, but has spread to several other products such as cosmetics [3]. Furthermore, as the world's fastest growing religion, which promotes hygiene and a high quality of products, halal certified cosmetics have a wider market appeal among non-Muslim consumers, who attribute these products with ethical consumerism and more stringent quality assurance standards [4].

As stated by Houlis in 2015, the market size for halal cosmetics was estimated at 20 billion USD, and it is expected to reach 54 billion USD by 2022 [5]. However, this volume is believed to represent only 2.5% of the global cosmetics industry [6]. Although Muslims

make up 25% of the world's population, the consumption of halal cosmetics is considerably low among Muslim consumers.

There are three categories of cosmetics products available in the market. These are:

(a) Halal cosmetics. These are products which have received a halal certification awarded by an authoritative body.

(b) Non-halal cosmetics. These are products defined as non-halal due to forbidden ingredients in their composition.

(c) Non-halal certified cosmetics. These are cosmetics which are free from forbidden ingredients in their composition, but whose manufacturers did not apply for a halal certification.

Although Muslims understand that halal products are confirmed pure, hygienic, and safe for consumption, most Muslim consumers remain loyal to non-halal certified cosmetics. In Malaysia, non-halal certified cosmetic brands such as SK-II, Chanel or Christian Dior are still extremely popular and frequently consumed by Muslims. Despite the absence of a halal certification, international cosmetics products are still popular among Muslim consumers [7]. Young consumers prefer non-halal certified cosmetics, and they are willing to pay extra to enjoy their perceived quality [8] in spite of the lack of compliance with the halal requirements [4]. Moreover, the addition of chemical compounds in cosmetics, [9] potentially creates a health threat [10]. According to Bilal, Mehmood, and Iqbal [9], from more than 12,000 industrial and synthetic chemical ingredients included in cosmetic formulation, less than 20% have been considered safe for use. Adding to that Łodyga-Chruścińska, Sykuła, and Więdłocha [10] argued that excessive amounts of nickel and copper found in cosmetics could cause allergic reactions. This corroborates arguments brought forward by Sugibayashi et al. [4] that such products may not meet halal requirements, thus, justifying why most cosmetics manufacturers still lack halal certifications. The loyalty of Muslim consumers toward non-halal certified cosmetics has motivated us to explore the factors influencing the intention to use halal cosmetics among millennials in Malaysia. A preliminary interview with 15 university students, revealed that they were conscious that not all their cosmetics products were certified as halal. This indicates that despite being aware of the availability of halal cosmetics in the market, they still consumed non-halal certified cosmetics.

Several scholars explore the field of halal research. Their studies address issues of halal logistics [1], halal transportation [11], halal warehousing [12], halal food certification [13], and halal cosmetics [14]. However, none of them have compared males' and females' intention to use halal cosmetics. According to Marketing Chart [15], the consumption of cosmetics is no longer bound by gender, as males have also started using cosmetics daily. In America, on an average men and women respectively use six and twelve different cosmetics products on a daily basis [10]. Furthermore, the purchase intention of halal cosmetics among young consumers still remains underexplored [16].

The findings of the present study contribute to the body of knowledge in cosmetics, especially in the halal cosmetics literature. Filling the gap in the literature is achieved by extending the theory of planned behavior (TPB) created by Ajzen [17] with brand image, and by comparing between males and females. Furthermore, as developed by Fishbein and Ajzen [18] and by Ngah et al. [19], adding new variables in the theory yields a higher explanatory power. Hence, the authors included brand image as a new variable to extend the theory. Brand image is commonly associated with product quality and non-halal certified cosmetics already long established in the market have a good brand image. This potentially contributes significantly to all consumers' purchase decision, including Muslim consumers. Thus, the authors wanted to explore how brand image affects the decision to purchase halal cosmetics.

Owing to the scarcity of literature comparing males' and females' cosmetics consumption, especially of the halal kind, the findings of the study provide a new direction to promote halal cosmetics. Currently, in contradiction with Islamic teachings, most cosmetic

advertisements feature sensual female models. Hence findings from this study could influence new promotional methods aligned with shariah requirements.

### 1.1. Theory of Planned Behavior

The theory of planned behavior (TPB), developed by Ajzen [17] to enhance the theory of reasoned action, is used to predict consumer behavior on the intention to use and reuse. It consists of attitude, subjective norms (SN), and perceived behavioral control (PBC). The capability of the theory of planned behavior to explain individual future behavior is one of the reasons why this theory is widely utilized in consumer behavioral studies. Studies on halal certified products [13], regarding Muslims' willingness to pay for halal transportation [19] and whistle blowing intention among auditors [20] are based on this theory. This resulted in motivating the authors to employ this theory as the theoretical lens of the study. Moreover, few reports indicate that this theory was selected to research the field of halal cosmetics. By extending the theory with brand image, and comparing female and male students, the study enhances the capability of the TPB to predict future consumer behavior in halal cosmetics studies.

### 1.2. The Relationship between Attitude and the Intention to Purchase Halal Cosmetics

Attitude refers to an individual's feelings and perception influencing the future behavior [17]. Because of the strong connection between attitude and individual future behavior, attitude has been used regularly to predict consumer behavior across several areas of study. An individual having a positive attitude toward a particular behavior creates a positive intention toward that behavior. Numerous studies demonstrate that attitude has a positive relationship with intention to purchase halal products. Attitude is found to have a positive effect on the purchase intention of halal cosmetics [21–23]. Postulated to that, the authors propose that:

**Hypothesis 1 (H1a).** *Attitude has a positive influence on the intention to purchase halal cosmetics for all students.*

**Hypothesis 1 (H1b).** *Attitude has a positive influence on the intention to purchase halal cosmetics for female students.*

**Hypothesis 1 (H1c).** *Attitude has a positive influence on the intention to purchase halal cosmetics for male students.*

### 1.3. The Relationship between Subjective Norms and the Intention to Purchase Halal Cosmetics

Subjective norms (SN) refer to the specific behavior of important individuals resulting in the perceived social pressure to comply. For the context of this study, subjective norms refer to family members or individuals who are important to students in regard to purchasing halal cosmetics. Most students still shop with friends and family members. Hence, the authors would like to explore how these people who are important in students' lives could influence their purchasing behavior of halal cosmetics. Recent studies [24,25] revealed that SN have a positive relationship with the intention to purchase halal products. Postulated to that, the authors propose that:

**Hypothesis 2 (H2a).** *Subjective norms have a positive influence on the intention to purchase halal cosmetics for all students.*

**Hypothesis 2 (H2b).** *Subjective norms have a positive influence on the intention to purchase halal cosmetics for female students.*

**Hypothesis 2 (H2c).** *Subjective norms have a positive influence on the intention to purchase halal cosmetics for male students.*

### 1.4. The Relationship between Perceived Behavioral Control and the Intention to Purchase Halal Cosmetics

Perceived behavioral control (PBC) is regarded as the judgement by individuals on the ease or difficulty to perform a behavior. As aligned to the context of the present study, it is the students' perception of whether they have control over their behavior to purchase halal cosmetics. It can be viewed from the perspective of availability of resources, such as monetary or the availability of halal cosmetics where they shop. Previous studies [22,24,25] confirm that the PBC has a positive relationship with the intention to purchase halal products. Hence, the authors propose that:

**Hypothesis 3 (H3a).** *Perceived behavioral control has a positive influence on the intention to purchase halal cosmetics for all students.*

**Hypothesis 3 (H3b).** *Perceived behavioral control has a positive influence on the intention to purchase halal cosmetics for female students.*

**Hypothesis 3 (H3c).** *Perceived behavioral control has a positive influence on the intention to purchase halal cosmetics for male students.*

### 1.5. The Relationship between Brand Image and the Intention to Purchase Halal Cosmetics

Brand image refers to the image of a brand name, signs, symbols, and designs or the accumulation of these that represent the goods and differentiates them from their competitors [26]. Brand image is equally important for the products since it differentiates particular products from others [27]. For a product's brand image to influence purchasing behavior, consumers must be aware of its availability in the market. Positive image of the brand could favor the decision to purchase a product especially when consumers have abundant choices. Hence, having a positive brand image could be a favorable competitive advantage for a product.

A previous study conducted by Sanny et al. [28] revealed that brand image has a positive relationship with purchasing cosmetic products. Other studies [29,30] also highlighted the positive influence of brand image toward purchase intention. Postulated to that, the authors propose that:

**Hypothesis 4 (H4a).** *Brand image has a positive influence on the intention to purchase halal cosmetics for all students.*

**Hypothesis 4 (H4b).** *Brand image has a positive influence on the intention to purchase halal cosmetics for female students.*

**Hypothesis 4 (H4c).** *Brand image has a positive influence on the intention to purchase halal cosmetics for male students.*

### 1.6. The Role of Gender in Buying Behavior

Originally, cosmetic products were created to fulfil the demands of women, owing to the understanding that most cosmetics manufacturers offer feminine and beautifying products such as make-up [28]. Hence, fragrance, skin care, lipstick, and other cosmetics are designed for women. As can be observed, most cosmetics advertisements employ women as models. However, manufacturers recently started producing cosmetics for men to enhance boyish, charming, and younger looks [28]. Mary Kay, Clinique, Calvin Klein are examples of leaders in cosmetics producing products for both groups of consumers. As confirmed by Marketing Chart [15], cosmetics are no longer only for women, as men are also concerned about their appearance. To meet clients' demand during meetings or to be impressive during dates, 64.4% of men started to consume cosmetics products [31].

This shows that improving confidence levels by enhancing one's looks with cosmetics is not only for women but also for men.

Although cosmetics products are created for men and women, due to various reasons, the range of products for women are more than those for men. Moreover, the number of manufacturers focusing on the female clientele is higher than those dedicated to males. Furthermore, even if males also use cosmetics products, their usage and functions remain less significant than females. Males are more independent while choosing products, whereas females commonly refer to their family members or colleagues while deciding to buy new products. Because of that, certain scenarios for the relationship between predictors and criterion variables will differ for males and females. Figure 1 illustrates the research framework of the study. Based on this argument, the authors propose that:

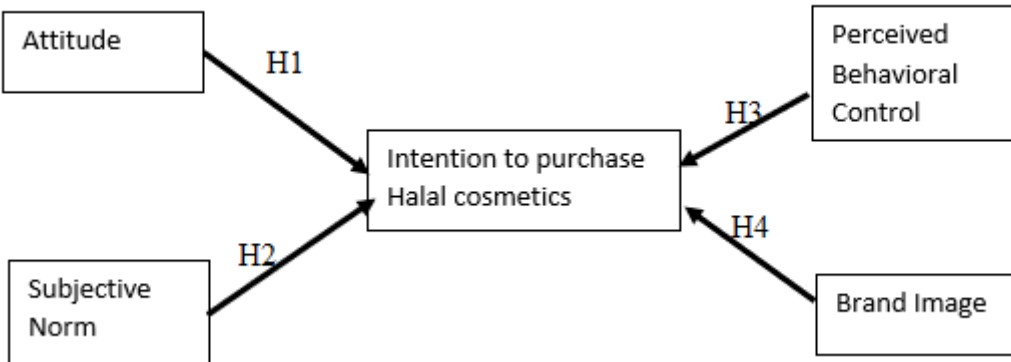

**Figure 1.** Research framework.

**Hypothesis 5 (H5a).** *The relationship between attitude and the intention to purchase halal products will be stronger for females compared to males.*

**Hypothesis 5 (H5b).** *The relationship between subjective norms and the intention to purchase halal cosmetics will be stronger for females compared to males.*

**Hypothesis 5 (H5c).** *The relationship between perceived behavioral control and the intention to purchase halal cosmetics will be stronger for males compared to females.*

**Hypothesis 5 (H5d).** *The relationship between brand image and the intention to purchase halal cosmetics will be stronger for females compared to males.*

## 2. Methods

The data were gathered from three universities in Malaysia. Since the study relates to the notion of halal, the sample was taken from universities offering Islamic studies programs. Using a purposive sampling method, Muslim students with at least a degree and who were mostly over the age of 21 years old were selected for their maturity to decide on their personnel purchasing behavior. Thus, students at the diploma level were excluded from the study. Data were collected via a survey. All respondents were approached while having their meals at cafeterias or while hanging out with their classmates around the universities. Respondents were asked about their willingness to participate in the study prior to receiving the questionnaire. Two filter questions formulated as "I am aware of the availability of halal cosmetics on the market" and "Currently, I am using non-halal certified cosmetics products" were inserted in the questionnaire to ensure the respondents' validity.

All the instrument constructs were adopted from established sources in the area of study. Attitude and perceived behavioral control were adopted from Ngah et al. [19], subjective norm from Iranmanesh et al. [13], brand image from Lee and Lee [32], and reuse intention from Venkatesh et al. [33]. From more than 700 students who were approached,

the researchers managed to collect answers from 578 respondents. However, after screening and sorting, 13.3% (77) of responses were discarded. Invalid respondents accounted for 41 questionnaires, and 36 respondents were discarded for answering in a straight line or due to missing values. Therefore, data from 501 respondents were used to analyze the relationship between variables in the study. Exactly 55.5% were 24 years old and above. Women represented 64.1% of the respondents. All respondents had a bachelor's degree level and 56.7% were from Islamic study programs. According to the year of studies, 46.7% of the respondents were 3rd year students, followed by 2nd year (28.5%), 1st year (16.2%), and final year students (8.6%). A total of 85% of respondents claimed that at the time of the study they consumed international brand cosmetics.

## 3. Results

Structural equation modelling with Smart PLS (v. 3.3.3) [34] was employed to test the hypotheses of the study. The Smart PLS (v. 3.3.3) software is well suited to the predictive nature of the study [35].

### 3.1. Common Method Variance

Prior to further analysis, common method variance (CMV) needed to be addressed to ensure that the results of the study were free from bias. Because of the single source of the data, issues of CMV could have been encountered [36]. As proposed by Mackenzie and Podsakoff [37], CMV can be remedied using procedural and statistical methods. The procedural method was addressed using different anchor scales to measure exogenous and endogenous variables. To confirm that CMV was not severe in the present study, a full collinearity analysis as proposed by Kock [38] was conducted. Accordingly, if the variance inflated factor (VIF) is greater than or equal to 3.3, then the study suffers from CMV [38]. Table 1 demonstrates that the CMV was not severe as all the VIF values were lower than the threshold value mentioned in the literature.

**Table 1.** Full collinearity.

| Group | ATT | Brand | PBC | Reuse Intention | SN |
|---|---|---|---|---|---|
| Female | 1.610 | 1.127 | 1.191 | 1.576 | 1.388 |
| Male | 1.632 | 1.283 | 1.434 | 1.727 | 1.368 |

### 3.2. Measurement Model

Following guidelines related to structural equation modelling analysis, the authors used a two-step approach to test the measurement model and the structural model. The measurement model tested items to ensure that when used for a specific variable, they truly measured that variable and it differed from others. Consequently, two types of validities needed to be achieved. These were the convergent validity and the discriminant validity. According to Hair et al. [36], convergent validity is confirmed if the loading is greater than or equal to 0.708, average variance explained (AVE) is greater than or equal to 0.5, and the composite reliability (CR) is greater than or equal to 0.7. Table 2 illustrates the results of the convergent validity testing for overall data, and both male and female groups. All values for loading, AVE and CR were higher than their threshold values, thus confirming that the convergent validity of the study was established. Appendix A provides the measurement items of the study.

**Table 2.** Convergent validity.

| Group | Item | Loading (Overall) | CR | AVE | Loading (F) | CR | AVE | Loading (M) | CR | AVE |
|---|---|---|---|---|---|---|---|---|---|---|
| Attitude | Att1 | 0.839 | 0.925 | 0.754 | 0.841 | 0.926 | 0.758 | 0.835 | 0.923 | 0.750 |
| | Att2 | 0.888 | | | 0.887 | | | 0.889 | | |
| | Att3 | 0.914 | | | 0.921 | | | 0.901 | | |
| | Att4 | 0.831 | | | 0.832 | | | 0.837 | | |
| Brand | Brand1 | 0.877 | 0.903 | 0.757 | 0.899 | 0.903 | 0.756 | 0.849 | 0.903 | 0.757 |
| | Brand2 | 0.887 | | | 0.896 | | | 0.877 | | |
| | Brand3 | 0.844 | | | 0.810 | | | 0.883 | | |
| Intention | Int1 | 0.917 | 0.929 | 0.814 | 0.917 | 0.915 | 0.783 | 0.920 | 0.953 | 0.872 |
| | Int2 | 0.923 | | | 0.916 | | | 0.941 | | |
| | Int3 | 0.867 | | | 0.817 | | | 0.941 | | |
| PBC | Pbc1 | 0.843 | 0.867 | 0.620 | 0.892 | 0.856 | 0.599 | 0.819 | 0.876 | 0.639 |
| | Pbc2 | 0.779 | | | 0.717 | | | 0.816 | | |
| | Pbc3 | 0.762 | | | 0.741 | | | 0.780 | | |
| | Pbc4 | 0.762 | | | 0.734 | | | 0.782 | | |
| SN | Sn1 | 0.830 | 0.893 | 0.676 | 0.847 | 0.883 | 0.655 | 0.790 | 0.913 | 0.724 |
| | Sn2 | 0.845 | | | 0.821 | | | 0.890 | | |
| | Sn3 | 0.828 | | | 0.804 | | | 0.877 | | |
| | Sn4 | 0.785 | | | 0.762 | | | 0.844 | | |

As proposed by Franke and Sarstedt [39] to establish discriminant validity, researchers should rely on the hetero-trait mono-trait (HTMT) ratio. Discriminant validity is confirmed when all values for HTMT are smaller than or equal to 0.85. Table 3 shows the results for the HTMT for overall data, female and male groups. All the HTMT values were lower than 0.85, thus establishing the discriminant validity of the study.

**Table 3.** Discriminant validity (hetero-trait mono-trait (HTMT)).

| Group | Construct | ATT | Brand | PBC | Intention | SN |
|---|---|---|---|---|---|---|
| Overall | ATT | | | | | |
| | Brand | 0.287 | | | | |
| | PBC | 0.249 | 0.327 | | | |
| | Intention | 0.608 | 0.349 | 0.344 | | |
| | SN | 0.495 | 0.227 | 0.416 | 0.409 | |
| Female | ATT | | | | | |
| | Brand | 0.226 | | | | |
| | PBC | 0.262 | 0.337 | | | |
| | Intention | 0.639 | 0.284 | 0.263 | | |
| | SN | 0.508 | 0.248 | 0.396 | 0.468 | |
| Male | ATT | | | | | |
| | Brand | 0.416 | | | | |
| | PBC | 0.228 | 0.308 | | | |
| | intention | 0.561 | 0.460 | 0.477 | | |
| | SN | 0.470 | 0.195 | 0.452 | 0.317 | |

*3.3. Measurement Invariance of Composites*

To compare the findings between males and females on the intention to purchase halal cosmetics, the measurement invariance of composites (MICOM) must be confirmed [36]. In response to the limitation of the previous method by co-variance-based SEM, the measurement invariance of composite was developed by Henseler, Ringle, and Sarstedt [40]. This method applies to PLS-SEM. However, prior to testing the findings between groups, a three step-process is required for the MICOM model. These are: (1) configural invariance assessment, (2) compositional invariance assessment, and (3) equal means and variances must be fulfilled. If the study passes all requirements in step two and three, it is possible to test the hypotheses for the overall, female and male groups. For the first step, the configural invariance was established since all data have an identical indicator, an identical data treatment, and an identical algorithm setting. The results of compositional

invariance and equal means, as presented in Table 4, reveal that steps two and three of the MICOM model were established. Consequently, the data enabled the hypotheses testing for the overall, female and male groups in the current study.

**Table 4.** The measurement invariance of composites (MICOM).

| Group | Construct | Original Correlation | Confidence Interval | Partial Measurement Invariance |
|---|---|---|---|---|
| STEP 2 | ATT | 1 | 0.998;1.000 | YES |
| | Intention | 0.999 | 0.999;1.000 | YES |
| | Brand | 0.997 | 0.994;1.000 | YES |
| | PBC | 0.975 | 0.968;1.000 | YES |
| | SN | 0.997 | 0.990;1.000 | YES |
| STEP 3 | **Construct** | **Variance** | **Confidence Interval** | **Full Measurement Invariance** |
| | ATT | −0.251 | −0.533; 0.49 | YES |
| | Intention | 0.124 | −0.398; 0.363 | YES |
| | Brand | −0.058 | −0.329; 0.303 | YES |
| | PBC | 0.028 | −0.334; 0.312 | YES |
| | SN | −0.084 | −0.25; 0.257 | YES |

*3.4. Structural Model*

Before proceeding with the hypotheses testing, it was crucial for the authors to ensure that the multi-collinearity of the study would not affect the findings. As proposed by Diamantopoulos and Siguaw [41], multi-collinearity does not affect the quality of the findings if the VIF value is smaller than or equal to 3.3. Since all the VIF values for each hypothesis were smaller than or equal to 3.3, it confirms that multi-collinearity was not a serious issue in the study.

To test the hypotheses of the study, a bootstrapping technique with a 5000 resampling procedure was applied. Hypotheses are supported if the beta value aligns with the direction of the hypothesis, the t-value is greater than or equal to 1.645, the *p*-value is smaller than or equal to 0.05, and there is no zero in straddle between the lower level (LL) and upper level (UL) of confidence intervals. Table 5 and Figure 2 illustrate the results of hypotheses testing of the study consisting of full and split datasets. Hypotheses 1a–4a relate to overall students, hypotheses 1b–4b relate to female students, whereas hypotheses 1c–4c deal with male students. Interestingly, all hypotheses for the overall data were found supported, findings for female and male students resulted in only three supported hypotheses.

**Table 5.** Hypotheses testing.

| Hypothesis | Relationship | Beta | SE | T Value | *p* Values | LL | UL | R2 | F2 | VIF |
|---|---|---|---|---|---|---|---|---|---|---|
| H1a | ATT -> Int | 0.443 | 0.042 | 10.110 | 0.001 | 0.373 | 0.512 | 0.359 | 0.239 | 1.284 |
| H2a | SN -> Int | 0.087 | 0.043 | 1.963 | 0.023 | 0.017 | 0.163 | | | 1.344 |
| H3a | PBC -> Int | 0.145 | 0.038 | 3.800 | 0.001 | 0.082 | 0.208 | | 0.027 | 1.201 |
| H4a | Brand -> Int | 0.137 | 0.044 | 3.302 | 0.001 | 0.070 | 0.210 | | 0.026 | 1.136 |
| H1b | ATT -> Int | 0.462 | 0.059 | 8.114 | 0.001 | 0.366 | 0.562 | 0.366 | 0.264 | 1.274 |
| H2b | SN -> Int | 0.149 | 0.059 | 2.669 | 0.006 | 0.058 | 0.251 | | 0.026 | 1.353 |
| H3b | PBC -> Int | 0.072 | 0.046 | 1.593 | 0.059 | 0.000 | 0.150 | | | 1.183 |
| H4b | Brand -> Int | 0.105 | 0.054 | 2.104 | 0.025 | 0.032 | 0.204 | | 0.020 | 1.110 |
| H1c | ATT -> Int | 0.403 | 0.069 | 6.073 | 0.001 | 0.267 | 0.501 | 0.417 | 0.198 | 1.362 |
| H2c | SN -> Int | −0.035 | 0.055 | 0.711 | 0.262 | −0.128 | 0.051 | | | 1.366 |
| H3c | PBC -> Int | 0.327 | 0.065 | 4.994 | 0.001 | 0.226 | 0.434 | | 0.148 | 1.249 |
| H4c | Brand -> Int | 0.184 | 0.075 | 2.439 | 0.007 | 0.057 | 0.309 | | 0.048 | 1.225 |

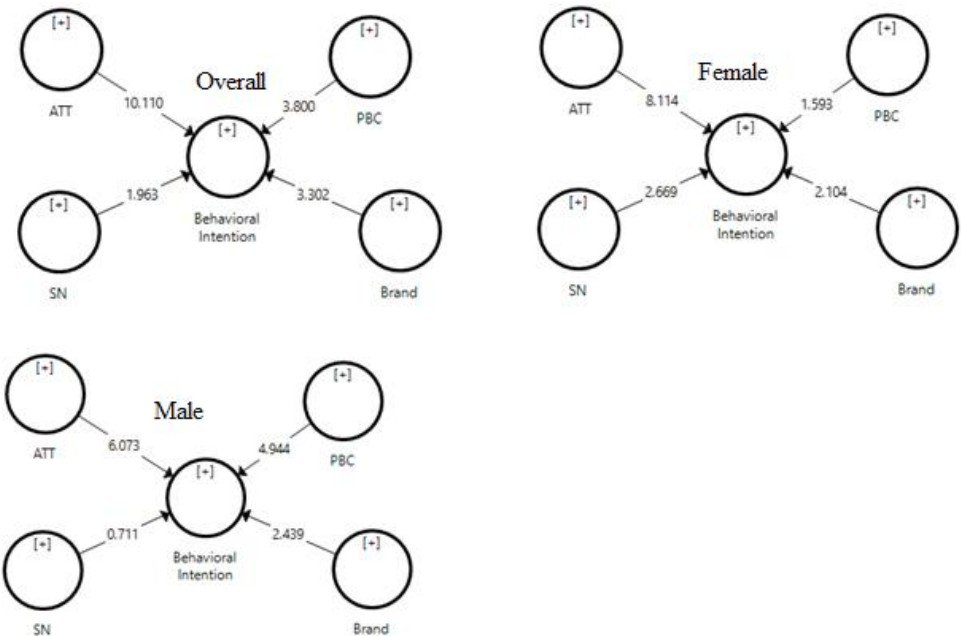

**Figure 2.** Structural model for overall, female and male groups.

For hypothesis 1, for overall student data, female and male students, the results reveal that attitude has a positive effect on the intention to purchase halal cosmetics, thus, supporting H1a (β = 0.444, $p \leq 0.01$), H1b (β = 0.461, $p \leq 0.01$), and H1c (β = 0.403, $p \leq 0.01$). For hypothesis 2, overall students (β = 0.087, $p \leq 0.05$) and female students (β = 0.149, $p \leq 0.01$), the results reveal that subjective norms have a positive effect on these two groups' intention to purchase halal cosmetics. Conversely, for male students (β = −0.035, $p = 0.262$), the analysis reveals that subjective norms have no effect on their intention to purchase halal cosmetics. Thus, the results demonstrate that H2a and H2b were supported, whereas H2c was unsupported. For hypothesis 3, the results show that PBC has a positive effect on the overall student group (β = 0.145, $p \leq 0.01$) and on male students (β = 0.327, $p \leq 0.01$), thus supporting H3a and H3c. However, the results also demonstrate that PBC has no effect on the intention to purchase halal cosmetics for female students; thus, H3b of the study is unsupported. For the last direct hypothesis, all groups of data confirmed that brand image has a positive effect on the intention to purchase halal cosmetics, thus supporting H4a (β = 0.137, $p \leq 0.01$), H4b (β = 0.443, $p \leq 0.01$), and H4c (β = 0.184, $p \leq 0.01$).

For the explained variance ($R^2$), the results show that the four independent variables of the study explain 36% of the variance in the intention to purchase halal cosmetics for the full dataset, 36.5% of the variance in the female dataset and 42.1% of the variance in the male dataset. For the $f^2$, as Cohen [42] proposed 0.02, 0.15, and 0.35 as small, medium, and large effect size respectively, the analysis revealed that attitude had a medium effect size for all three datasets. Furthermore, attitude had the strongest effect size of all variables tested. Table 6 shows the results for the $R^2$ and $f^2$ values of the study.

**Table 6.** Hypotheses testing for multigroup analysis.

| | | MGA | | Welch-Satterthwait Test | | |
|---|---|---|---|---|---|---|
| **Hypothesis** | **Relationship** | **Beta** | ***p*-Value** | **t-Value** | ***p*-Value** | **Decision** |
| H5a | ATT -> Behavioral Intention | 0.058 | 0.262 | 0.642 | 0.23 | Unsupported |
| H5b | SN -> Behavioral Intention | 0.189 | 0.011 | 2.337 | 0.012 | Supported |
| H5c | PBC -> Behavioral Intention | −0.265 | 0.001 | 3.356 | 0.001 | Supported |
| H5d | Brand -> Behavioral Intention | −0.059 | 0.256 | 0.686 | 0.196 | Unsupported |

*3.5. Assessment of Group Differences*

PLS multi-group analysis (MGA) was conducted to test the differences between the female and male datasets. The results were based on the MGA and Welch-Satterthwait Test. Based on MGA [43], the hypothesis is supported if the p-value is less than 0.5 or higher than 0.95. According to Sarstedt, Henseler, and Ringle [44], in the Welch-Satterthwait Test, the hypothesis is supported if the t-value is higher than 1.96 and the *p*-value is lower than 0.95. Table 6 shows the analysis of the multi-group analysis which reveals that two hypotheses were supported, and another two were unsupported. For H5a and H5d, the results indicate that there is no significant difference for the relationship between attitude and intention (H5a) and brand image and intention to purchase halal products, thus H5a and H5d were identified as unsupported. Whereas, for H5b, the results indicate that there is a significant difference for the relationship between female and male students, where the relationship is stronger for females compared to the males, thus supporting H5b. Likewise for H5c, the results revealed that there was a significant difference between female and male students in the relationship between PBC and the intention to purchase halal products. The negative beta indicated a weaker relationship for female compared to male, hence H5c was supported.

**4. Discussion**

To predict future behavior of Muslim consumers' intention to purchase halal cosmetics, the authors employed the TPB with the extension of brand image. Although cosmetics are commonly associated with women, men have started using various types of cosmetics products to accentuate their personality and confidence level in their daily routines.

The analysis reveals that attitude has a positive relationship with the intention to purchase halal cosmetics for the overall, female and male datasets, thus corroborating the findings from previous studies [21–23] which highlighted that attitude has a positive effect on the intention to purchase halal cosmetics. Currently, respondents still consume non-halal certified products. By having a positive attitude and improving some factors of halal cosmetics manufacturers, the future may see them switch to halal cosmetics. Thus, in order to increase the likelihood of this occurring, halal cosmetics manufacturers should create positive attitudes among Muslim consumers toward their products. Linking their products to respected figures and working with appropriate models to promote their products could create positive attitudes among Muslim consumers.

For the subjective norms, mixed results were obtained. Results reveal that for the overall and female datasets, the subjective norms have a positive effect on the intention to purchase halal cosmetics. These results are aligned with the previous studies [24,25] which demonstrate that subjective norms have a positive relationship with the intention to purchase halal cosmetics. However, for the male dataset, the results show that the hypothesis was unsupported. These findings are aligned with other studies in the field of halal matters such as the one by Soon and Wallace [45]. Even though males are starting to use cosmetics products, not all of them have enough knowledge on such products. Furthermore, it is common for the younger generation to decide what is best for them and not to be influenced by others in their decisions, especially for males on issues of cosmetics.

Perceived behavioral control also revealed different results for females and males. The analysis shows that perceived behavioral control has a positive effect in the overall and

male datasets. The findings strengthen previous findings from several researchers [22,25] who found that perceived behavioral control has a positive relationship with the intention to purchase halal products. However, the analysis shows that, perceived behavioral control has no effect in the female dataset. The findings are aligned with Ali et al. [24], who noted that perceived behavioral control is not a significant factor for the low materialism group to purchase halal products. The logical reason for this finding is that once females find products that suit them, they try their best to obtain these products. Furthermore, since consuming halal products is not an option but a responsibility for Muslims, the ease or difficulty of obtaining these products is no longer an issue.

Lastly, brand image was found to have a positive effect on the intention to purchase halal cosmetics for all datasets of the study, hence confirming the essentiality of brand image toward the intention to purchase halal products. A good brand image could be developed if halal cosmetics manufacturers sponsor major Islamic program events which are aired on television, radio, and other social media channels. Furthermore, as Islam is the official religion of Malaysia, several Islamic programs are available and followed by Muslim consumers. Associating with these programs could create a better brand image for Muslim consumers in Malaysia.

For the multigroup analysis, the results show that there are significant differences between the female and male datasets. Subjective norms were found to have a stronger effect on females, whereas perceived behavioral control was found to have a stronger effect on males. Understandably, the respondents of the study are still using non-halal certified cosmetics from international brands. Most international brands are relatively pricey, and since the respondents are students, purchasing expensive cosmetic remains an obstacle for them. This is particularly true for females; hence, the study demonstrates that the effect of subjective norms is stronger for females than for males.

For perceived behavioral control, the results show that the effect was stronger for males than for females. In effect, most men are unwilling to spend more time on shopping, especially for cosmetics. Hence, the ease or difficulty to obtain a product has an impact on their purchasing behavior. Compared to females, males view shopping as a sort of treatment, specially to purchase cosmetics products. The ease or difficulty is not an issue if they are willing and content to get the products. For this reason, perceived behavioral control has a greater impact on male than on female students.

For attitude and brand image, the results establish that there is no significant difference between female and male students. It shows that even if they have different characteristics and different purposes for using the products, the effect of attitude and brand image of cosmetics products are not different for them. To increase the intention to purchasing halal cosmetics, halal cosmetics manufacturers should not differentiate between female and male Muslim consumers in their effort to promote positive attitude and brand image.

## 5. Conclusions

The Muslim consumer market is rapidly growing, driven by an increasing population that is more diverse ethnically, geographically, and economically than ever before. Young and product-conscious Muslims are driving the demand for halal cosmetics, with the market growing from an estimated 20 billion USD in 2015 to an expected 54 billion USD by 2022. This shows the huge potential of halal cosmetics to meet the world demand for Muslim and non-Muslim consumers. As halal food is widely accepted, the same should apply to cosmetics.

From the analysis, the authors conclude that attitude, subjective norms, perceived behavioral control and brand image have a positive impact on the intention to purchase halal cosmetics. However, their effects differ between female and male consumers. Hence, halal cosmetics manufacturers should be keen to digest the findings of the study to craft better marketing approaches and strategies to increase the acceptance of halal cosmetics for both female and male Muslim students. Associating the products and brands with

respected figures in the country and sponsoring established religious programs through various channels should also be considered.

The study also contributes theoretically. The findings confirm the capability of the TPB, with the extension of brand image, to explain Muslim students' behavior toward halal cosmetics. By comparing females and males, the study also enriches the halal literature, particularly the halal cosmetics literature. Since the study is limited to students in three different universities in Malaysia, future studies should compare students with non-students. Alternatively, a future study could focus on educated respondents, as these are more knowledgeable and financially able to purchase what they prefer for their daily lives. Because the present study was limited to four predictor variables, future studies could explore why Muslim consumers remain loyal to non-halal certified cosmetics despite being obliged to use halal products. Moreover, as the study sample was limited to Malaysian respondents, the findings may not be generalized to the world's population. These might be applicable to countries with similar characteristics to Malaysia, especially those with a Muslim population aware of the concept of halal, and with a similar culture and lifestyle. Future studies should attempt to compare the intention to purchase halal cosmetics across different countries, especially in Muslim countries. Furthermore, future studies should also focus on specific products such as make-up, skin care, and other hygienic products.

**Author Contributions:** Conceptualization, A.H.N. and H.H.; methodology, A.H.N. and J.A.A.-G.; software, S.G.; validation, A.H.N., S.R., and H.H.; formal analysis, A.H.N.; investigation, S.-h.P.; resources, S.G., S.-h.P.; data curation, A.H.N.; writing—original draft preparation, A.H.N.; writing—review and editing, A.H.N., S.G., H.H. All authors have read and agreed to the published version of the manuscript.

**Funding:** This research received no external funding.

**Institutional Review Board Statement:** Not applicable.

**Informed Consent Statement:** Not applicable.

**Data Availability Statement:** Not applicable.

**Conflicts of Interest:** The authors declare no conflict of interest.

## Appendix A

| Measurement Item | Source |
| --- | --- |
| Attitude | Ngah et al. (2020) |
| Using for halal cosmetics is a good idea | |
| Using for halal cosmetics is a wise idea | |
| I like the idea of using halal cosmetics | |
| Using halal cosmetics is pleasant | |
| Subjective Norm | Iranmanesh et al (2019) |
| My family thinks that I should consume halal cosmetics rather than non-halal cosmetics | |
| Most people I value would consume halal cosmetics rather than non-halal cosmetics | |
| People I value (such as my friends) think I should consume halal cosmetics | |
| My close friends, whose opinions are important to me, think that I should consume halal cosmetics | |
| Perceived Behavioral Control | Ngah et al. (2020) |
| I have control to pay for halal transportation | |
| I have the resources necessary to pay for halal transportation | |
| I have the knowledge necessary to pay for halal transportation | |
| Given the resources, opportunities, and knowledge it would be | |
| Brand Image | Lee & Lee, 2018 |

| Measurement Item | Source |
| --- | --- |
| These brands play a leading role in the industry | |
| These brands' image is differentiated from other brands | |
| These brands are friendly | |
| Intention to use | (Venkatesh et al., 2003) |
| I intend to use the system in the next 3 months. | |
| I predict I would use the system in the next 3 months | |
| I plan to use the system in the next 3 months. | |

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
