# Peer review of "Intention to Purchase Halal Cosmetics: Do Males and Females Differ? A Multigroup Analysis"

_cosmetics, doi:10.3390/cosmetics8010019_

Round 1

Reviewer 1 Report

Abstract: ‘puzzling’ should be replaced with a more suited word to describe better the reported observation in context. Halal may not be italicized; it is an English word already.

‘Despite the acceptance of halal food, halal cosmetics remain scarce’ appears unwarranted. There are studies already about the purchasing behavior and acceptance of Muslims of halal cosmetics. Authors must clarify what is meant by ‘scarce’; do they mean the acceptance for halal cosmetics is low or the availability of halal cosmetics is scarce?

Introduction:

‘This explains why halal is related to Muslim, however the majority of halal certified manufacturers are non-Muslims’ It is already known that halal is associated with the Muslim population. This statement probably suggests that ‘halal products are central to Muslims yet the majority of manufacturers are non-Muslims’. There is no problem existing with ‘halal certified manufacturers’ with regards to halal products.

halal products have attracted non-Muslims, especially in cosmetics’ may be rephrased as ‘halal certified cosmetics a wider market appeal among non-Muslim consumers, who attribute these products with ethical consumerism and more stringent quality assurance standards’

‘Moreover, the addition of chemical compounds to enhance the quality of cosmetics creating a potential health threat corroborates arguments brought forward by that such products may not align with halal requirements resulting in the lack of halal certifications’ It is suggested that this sentence be simplified. Avoid using complex sentences.

Referring to ones’ work may be presented implicitly and not by in-text referencing using numbers in brackets.

The theoretical background may be presented in a more concise manner.

Methods:

The section heading may be changed to ‘Methods’

The inclusion and exclusion criteria, and location of the sampling site must be declared.

‘Moreover, it is believed that millennials are more self-reliant, image-driven, and emphasize more on materialistic values’ must be removed from the manuscript.

‘All the instrument constructs were adopted from established sources in the area of study’ insufficiently describes the contents of the instrument. What about categories of questions? How were the responses obtained (e.g., interview, email)? How was it recorded? Content analysis of responses?

Ethics review process considering the involvement of students as respondents.

Results: None

Discussion: None

Conclusion: None

Author Response

no

Comment

Action

Abstract: ‘puzzling’ should be replaced with a more suited word to describe better the reported observation in context. Halal may not be italicized; it is an English word already.

Thanks for your helpful comment

Puzzling was replaced with “trivial”

halal was change from italic to non-italic “halal’

‘Despite the acceptance of halal food, halal cosmetics remain scarce’ appears unwarranted. There are studies already about the purchasing behavior and acceptance of Muslims of halal cosmetics. Authors must clarify what is meant by ‘scarce’; do they mean the acceptance for halal cosmetics is low or the availability of halal cosmetics is scarce?

Thanks for your sharp view.

The sentence has been rephrased to “..literature on the acceptance of halal cosmetics remains scarce...”

‘This explains why halal is related to Muslim, however the majority of halal certified manufacturers are non-Muslims’ It is already known that halal is associated with the Muslim population. This statement probably suggests that ‘halal products are central to Muslims yet the majority of manufacturers are non-Muslims’. There is no problem existing with ‘halal certified manufacturers’ with regards to halal products.

Thanks a for a wise suggestion.

The text was changed as proposed.

halal products are central to Muslims yet the majority of manufacturers are non-Muslims’

halal products have attracted non-Muslims, especially in cosmetics’ may be rephrased as ‘halal certified cosmetics a wider market appeal among non-Muslim consumers, who attribute these products with ethical consumerism and more stringent quality assurance standards’

Thanks for a wise suggestion. Authors followed exactly as proposed.

halal certified cosmetics have a wider market appeal among non-Muslim consumers, who attribute these products with ethical consumerism and more stringent quality assurance standards.

‘Moreover, the addition of chemical compounds to enhance the quality of cosmetics creating a potential health threat corroborates arguments brought forward by that such products may not align with halal requirements resulting in the lack of halal certifications’ It is suggested that this sentence be simplified. Avoid using complex sentences.

This sentence has been reworded and modified. A few sentences were added to enhance the justification.

Moreover, the addition of chemical compounds in cosmetics, [9] potentially creating a potential health threat [10]. According to Bilal, Mehmood and Iqbal [9], from more than 12,000 industrial and synthetic chemical ingredients included in cosmetic formulation, less than 20% have been considered safe for use. Adding to that Łodyga-Chruścińska, Sykuła and Wiȩdłocha [10] argued that some metals found in cosmetics such as nickel and copper, cause allergic reactions.  This corroborates arguments brought forward by Sugibayashi et al. [4] that such products may not meet halal requirements. Thus, justifying why most cosmetics manufacturers still lack halal certifications.

The theoretical background may be presented in a more concise manner.

Thanks for the comment. This section has been improved accordingly. These sentences were added in the text.

The capability of the theory of planned behavior to explain individual future behavior is one of the reasons why this theory is widely utilized in consumer behavioral studies. Studies on halal certified products [13], regarding Muslims’ willingness to pay for halal transportation [19] and whistle blowing intention among auditors [20] are based on this theory. This resulted in motivating the authors to employ this theory as the theoretical lens of the study.

The section heading may be changed to ‘Methods’

Done

The inclusion and exclusion criteria, and location of the sampling site must be declared.

Using a purposive sampling method, Muslim students with at least a degree and who were mostly over the age of 21 years old were selected for their maturity to decide on their personnel purchasing behavior. Thus, students at the diploma level were excluded from the study. Data were collected via a survey. All respondents were approached while having their meals at cafeterias or while hanging out with their classmates around the universities. Respondents were asked about their willingness to participate in the study prior to receiving the questionnaire.

‘Moreover, it is believed that millennials are more self-reliant, image-driven, and emphasize more on materialistic values’ must be removed from the manuscript.

This statement has been removed from the text

‘All the instrument constructs were adopted from established sources in the area of study’ insufficiently describes the contents of the instrument. What about categories of questions? How were the responses obtained (e.g., interview, email)? How was it recorded? Content analysis of responses?

The instruments of the study will be enclosed in the appendix.

Ethics review process considering the involvement of students as respondents.

Ethics review processing is applicable in Malaysia, but not for this kind of study.

Reviewer 2 Report

Use of cosmetics among Muslim males. in my opinion, is not a topic worthy of conducting research on. Yes, men do use after-shave lotions, anti-perspirants, hair gels etc.; but I am not sure if I should consider them 'cosmetics'. But that's my personal view.

The paper needs some editing. The authors need to change expressions such as  "...developed by (18) ... [p.2 line 83] which does not make sense. 18 needs to be replaced by Fishbein and Ajzen. There are a few other expressions such as this one. 

is it necessary to use the expression "non-Halal certified'? Just non-halal should be enough.

In the hypotheses section authors have used '...overall students..' which needs to be changed to '... all students...'. 

On page 7, line 269 'Hypotheses are announced as supported...' the authors should drop '... announced as..'

The paper needs to be thoroughly revised.

Author Response

No

Comment

Response

1

Use of cosmetics among Muslim males. in my opinion, is not a topic worthy of conducting research on. Yes, men do use after-shave lotions, anti-perspirants, hair gels etc.; but I am not sure if I should consider them 'cosmetics'. But that's my personal view.

Thanks for the opinion. However, nowadays, males also start to enjoy cosmetics products in their daily live. This sentence has been added into the text to strengthen the worthiness to study males and females’ cosmetics purchasing behaviour.

 In America, the average men and women respectively use six and twelve different cosmetics products on a daily basis [10].

2

is it necessary to use the expression "non-Halal certified'? Just non-halal should be enough.

There are three categories of cosmetics products available in the market.  These are:

a) Halal cosmetics. These are products which have received a halal certification awarded by an authoritative body.

b) Non-halal cosmetics. These are product defined as non-halal due to forbidden in-gredients in their composition.

c) Non-halal certified cosmetics. These are cosmetics which are free from forbidden ingredients in their composition, but whose manufacturers did not apply for a halal certification.

3

The paper needs some editing. The authors need to change expressions such as  "...developed by (18) ... [p.2 line 83] which does not make sense. 18 needs to be replaced by Fishbein and Ajzen. There are a few other expressions such as this one. 

Relevant in-text citations have been amended to include the authors as the subject of the sentence where required.

4

In the hypotheses section authors have used '...overall students..' which needs to be changed to '... all students...'. 

Done

5

On page 7, line 269 'Hypotheses are announced as supported...' the authors should drop '... announced as..'

Done

6

The paper needs to be thoroughly revised.

The paper has been thoroughly revised.

Reviewer 3 Report

ref. line 48: it should be clarified that halal products are 'pure' only in the sense of being free of some forbidden ingredients. The concept of 'pure' in cosmetics is far more extensive, concerning heavy metals impurities, allergens, pesticides etc.

'hygienic' is a required quality for all cosmetic products in the world

'safe for consumption' is a general requirement of all cosmetics, not specific to halal cosmetics  

line 54: I would prefer the term 'perceived quality'

paragraph from 54 to 58 should be rewritten, it is not clear. If additives are inserted to improve quality, how could these ingredients pose a threat to health?  Cosmetics in general (halal or not halal) hould not be harmful for the consumers.

ref line 67: in this paper, it is evident a misconcept: cosmetics are not only make up products and creams, but also includes the whole category of hygiene products, from soap to shaving creams, from shampoos to sunscreens etc. In this sense they are not gender related and are used by both sexes according only their hygiene habits.

line 103 to 108: this subdivision could be summarized in just one phrase. The same applies for subdivisions at lines 118 to 12, and 133 to 138, 152 to 157.

line 159 Again this phrase could refer to creams and makeup, but it does not take into account at all body care products in ancient time, hair dyes in Asia and hygiene products in general. Makeup is a specific category, should not be generalised to all the cosmetic products. All the references to  cosmetics applied for appearance are wrong, as they  do not take into account the skin protection, healing hygiene, hair grooming aims etc. These statements should be rephrased or referred to specific categories of products.

line 168 169. Bibliography  refers to the Malaysian market. This detail  should be clearly identified. . At the light of this parameter, all the paper. should be clearly centered as a focus on the analysis of this specific national market segment, without making too much generalized statements.

all the descritpion form 245 to 320  are too much detailed and readable only by expert in statistics. please reduce and simplify.

final comments should report that in cosmetics the sensorial side is very important and could strongly influence puschasing

Author Response

No

Comment

Response

1

ref. line 48: it should be clarified that halal products are 'pure' only in the sense of being free of some forbidden ingredients. The concept of 'pure' in cosmetics is far more extensive, concerning heavy metals impurities, allergens, pesticides etc.

Thanks for your sharp view on the writing.

However, pure in the sentences regarding the purity of the Halal products is from an Islamic perspective.

Although Muslims understand that halal products are confirmed pure, hygienic and safe for consumption, most Muslim consumers remain loyal to non-halal certified cosmetics

2

'hygienic' is a required quality for all cosmetic products in the world.

'safe for consumption' is a general requirement of all cosmetics, not specific to halal cosmetics  

Thanks for the opinion.

Hygienic and safe for consumption should be the quality for all cosmetics products, however this as mentioned in the text, line 56-66

According to Bilal, Mehmood and Iqbal [9], from more than 12,000 industrial and synthetic chemical ingredients included in cosmetic formulation, less than 20% have been considered safe for use. Adding to that Łodyga-Chruścińska, Sykuła and Wiȩdłocha [10] argued that some metals found in cosmetics such as nickel and copper, cause allergic reactions.

3

line 54: I would prefer the term 'perceived quality'

Thanks for the suggestion,

The authors now use perceived quality in the text

4

paragraph from 54 to 58 should be rewritten, it is not clear. If additives are inserted to improve quality, how could these ingredients pose a threat to health?  Cosmetics in general (halal or not halal) hould not be harmful for the consumers.

Thanks for the good comment. The authors have reworded that sentence.

Moreover, the addition of chemical compounds in cosmetics, [9] potentially creating a potential health threat [10]. According to Bilal, Mehmood and Iqbal [9], from more than 12,000 industrial and synthetic chemical ingredients included in cosmetic formulation, less than 20% have been considered safe for use. Adding to that Łodyga-Chruścińska, Sykuła and Wiȩdłocha [10] argued that some metals found in cosmetics such as nickel and copper, cause allergic reactions. This corroborates arguments brought forward by Sugibayashi et al. [4] that such products may not meet halal requirements. Thus, justifying why most cosmetics manufacturers still lack halal certifications.

5

ref line 67: in this paper, it is evident a misconcept: cosmetics are not only make up products and creams, but also includes the whole category of hygiene products, from soap to shaving creams, from shampoos to sunscreens etc. In this sense they are not gender related and are used by both sexes according only their hygiene habits

Thanks for the helpful comments.

However, the study is not limited to cosmetics as make up and creams, but also includes several kinds of products. The study just gave an example based on the literature of types of cosmetics potentially harmful for some persons.

Adding to that Łodyga-Chruścińska, Sykuła and Wiȩdłocha [10] argued that some metals found in cosmetics such as nickel and copper, cause allergic reactions. This corroborates arguments brought forward by Sugibayashi et al. [4] that such products may not meet halal requirements. Thus, justifying why most cosmetics manufacturers still lack halal certifications. Allergic reactions do not refer to all consumers, but only to some persons.

6

line 103 to 108: this subdivision could be summarized in just one phrase. The same applies for subdivisions at lines 118 to 12, and 133 to 138, 152 to 157.

Thanks for the comments.

However, as far as authors concern, most of other authors in the top journals wrote the hypotheses as we did, if there are testing a multigroup analysis, as we did in the paper.

7

line 159 Again this phrase could refer to creams and makeup, but it does not take into account at all body care products in ancient time, hair dyes in Asia and hygiene products in general. Makeup is a specific category, should not be generalised to all the cosmetic products. All the references to  cosmetics applied for appearance are wrong, as they  do not take into account the skin protection, healing hygiene, hair grooming aims etc. These statements should be rephrased or referred to specific categories of products.

Thanks for a great view

Cream and make up also part of the cosmetics. The study used cosmetics in general term without focusing on the specific products. The authors highly appreciated the comments, and put this as a recommendation for future studies.

Future studies should attempt to compare the intention to purchase halal cosmetics across different countries, especially in Muslim countries. Furthermore, future studies should also focus on specific products such as make-up, skin care and other hygienic products.

8

line 168 169. Bibliography  refers to the Malaysian market. This detail  should be clearly identified. . At the light of this parameter, all the paper. should be clearly centered as a focus on the analysis of this specific national market segment, without making too much generalized statements.

Thanks for your concern. However, the study has no intention to over generalize the findings of the study to the world population. In the end of the paper, the study added new sentences to address this issue.

Moreover, as the study’s sample was limited to Malaysian respondents, the findings may not be generalized to the world’s population. These might be applicable to countries with similar characteristics to Malaysia, especially those with a Muslim population aware of the concept of halal, and with a similar culture and lifestyle.

9

all the descritpion form 245 to 320  are too much detailed and readable only by expert in statistics. please reduce and simplify

Thanks for your great concern.

However, all that is shown in the statistical analysis is based on the guidelines on reporting the findings of the study.

final comments should report that in cosmetics the sensorial side is very important and could strongly influence puschasing

The findings of the study are limited to the variables tested in the study. Since the purpose of the study is comparing the intention to purchase halal cosmetics based on the variables in the TPB and brand image, the discussion is only limited to these constructs. It is inadequate if the author discusses something which is beyond the scope, context, and objectives of the study.

Round 2

Reviewer 3 Report

Paper is OK, Nevertheless I insist (ref line 67) that metallic impurities cannot be avoided in cosmetics, provided that their amount is a few ppm and even Halal cosmetics cannot avoid them.

Author Response

Point 1: (x) English language and style are fine/minor spell check required

Thanks for your sharp view.

Response 1

The proof reader has made five changes; Line 52, 99, 118,199 and 287. The red font, is the previous version, green font is for the current version.

  • Product à Products (Line 52)

Original :

  1. b) Non-halal cosmetics. These are product defined as non-halal due to forbidden ingredients in their composition.

Latest : b) Non-halal cosmetics. These are products defined as non-halal due to forbidden ingredients in their composition. 

  • cosmetics advertisement à cosmetic advertisements (Line 99)

Original : Currently, in contradiction with Islamic teachings, most cosmetics advertisement feature sensual female models

Latest    : Currently, in contradiction with Islamic teachings, most cosmetic advertisements feature sensual female models

  • area of studies à areas of study (Line 118)

Original : Due to the strong connection between attitude and individual future behavior, attitude has been used regularly to predict consumer behavior across several area of studies

Latest    : Due to the strong connection between attitude and individual future behavior, attitude has been used regularly to predict consumer behavior across several areas of study.